**Data Availability Statement:** All relevant data are within the manuscript and its Supporting Information files.

# Cumulative burden of non-communicable diseases predicts COVID hospitalization among people with HIV: A one-year retrospective cohort study

**Michael D. Virata**[1]*, **Sheela V. Shenoi**[1], **Joseph Ladines-Lim**[2,3], **Merceditas S. Villanueva**[1], **Lydia A. Barakat**[1]

1 Department of Internal Medicine, Section of Infectious Diseases, Yale School of Medicine, New Haven, Connecticut, United States of America, 2 Department of Internal Medicine, University of Michigan, Ann Arbor, Michigan, United States of America, 3 Department of Pediatrics, University of Michigan, Ann Arbor, Michigan, United States of America

* michael.virata@yale.edu

## Abstract

There continue to be conflicting data regarding the outcomes of people with HIV (PWH) who have COVID-19 infection with most studies describing the early epidemic. We present a single site experience spanning a later timeframe from the first report on January 21, 2020 to January 20, 2021 and describe clinical outcomes and predictors of hospitalization among a cohort of PWH in an urban center in Connecticut, USA. Among 103 PWH with controlled HIV disease, hospitalization occurred in 33% and overall mortality was 1%. HIV associated factors (CD4 count, HIV viral suppression) were not associated with hospitalization. Chronic lung disease (OR: 3.35, 95% CI:1.28–8.72), and cardiovascular disease (OR: 3.4, 95% CI:1.27–9.12) were independently associated with hospitalization. An increasing number of non-communicable comorbidities increased the likelihood of hospitalization (OR: 1.61, 95% CI:1.22–2.13).

## Introduction

The first laboratory-confirmed case of coronavirus disease (COVID-19) secondary to severe acute respiratory syndrome coronavirus 2 (SARS-CoV-2) infection in the United States (US) was reported on January 21, 2020 [1]. A mere 12 months later, there were 24.5 million reported cases, with over 500,000 deaths [2].

Initial reports from China identified a number of risk factors for severe COVID-19 and mortality, including older age, male gender, immunosuppressed status, and co-morbidities such as hypertension, diabetes, chronic cardiovascular and respiratory disease [3–6]. Initially, it was predicted that HIV infection would be a risk factor for severe COVID-19 but studies have had mixed conclusions. The earliest case series showed that clinical outcomes among PWH were not any worse than those for patients without HIV [3, 7, 8] and found no excess risk of morbidity and mortality in symptomatic SARS-CoV-2 co-infected patients with fully

**Funding:** Sheela Shenoi received support from the Doris Duke Charitable Foundation (#201516). The funder did not play any role in the study design, data collection and analysis, decision to publish or preparation of the manuscript. All other authors received no specific funding for this work.

**Competing interests:** Sheela Shenoi's spouse worked for Merck Pharmaceuticals from 1997 to 2007 and retains company stock in his retirement account. While no conflict of interest is involved, this information is included here in the interest of full disclosure. All other authors report no potential conflicts of interest. This does not alter our adherence to PLOS ONE policies on sharing data and materials.

suppressed HIV, compared with HIV-uninfected patients [9, 10]. Similarly, a US study matched 21 HIV-infected against 42 HIV-uninfected patients, all hospitalized for COVID-19, and noted no significant difference in clinical course or outcomes [11]. A few months into the pandemic, there was more evidence from several areas in the US that HIV was not being identified as a risk factor for hospitalization [12]. However, a systematic review of 25 published studies of PWH co-infected with SARS-CoV-2 demonstrated that among 252 patients, 65% were hospitalized and 17% required admission to an intensive care unit (ICU) [13]. Other studies reported that PWH and COVID had worse outcomes with some suggesting that HIV-related factors such as CD4 count predicted poorer outcomes [14–16]. Many of these studies were from the earliest months of the pandemic and may not have reflected subsequent advances in COVID treatment and enhanced prevention efforts (e.g. masking and lockdowns). Given this conflicting data, we sought to evaluate whether PWH diagnosed with COVID-19 were at increased risk of hospitalization and characterize their hospital outcomes using data from the first year of the pandemic.

## Methods

### Study design and population

The study involved a retrospective chart review of PWH aged 18 years and older who had a laboratory-confirmed SARS-CoV-2 infection, defined with a positive reverse transcription polymerase chain reaction (RT-PCR) test result, between January 21, 2020, and January 20, 2021. All patients received their HIV care at two ambulatory clinics within the Yale–New Haven Hospital (YNHH) academic medical center in New Haven, Connecticut. The protocol was approved by the Yale University Institutional Review Board and fully complied with ethical practices. As this was a retrospective review of medical records, the requirement for written informed consent was waived. Data were anonymized prior to analysis.

### Data sources and analysis

A standardized form was used in abstracting data from electronic medical records (Epic Systems Corporation) including demographic information, co-morbidities, and clinical parameters such as body mass index (BMI), as well as HIV-specific data including AIDS diagnosis, antiretroviral therapy (ART), CD4 cell count and HIV viral load (HIVVL) prior to admission. Among the COVID-specific information items were COVID-19-related symptoms, reverse transcription polymerase chain reaction (RT-PCR) test results, and COVID-19 management protocols including specific SARS-CoV-2 therapies. Predetermined clinical outcomes included the need to escalate to ICU level of care, and all-cause mortality. HIVVL suppression was defined as <200 copies per milliliter (copies/ml) of blood; history of AIDS, as having had an opportunistic infection or a CD4 count of <200 cells/mm$^3$ or both; and obesity, as BMI>30.

Continuous and categorical variables were summarized as means or medians, and percentages as appropriate, and bivariate analysis was performed to compare hospitalized and non-hospitalized patients. Multivariable backward logistic regression identified independent correlates for hospitalization and calculated odds ratios (OR) with 95% confidence interval (CI). Differences with a p-value less than 0.05 were significant. The analysis used SPSS Version 26.0.

## Results

### Clinical characteristics of PWH with COVID-19

Among 1,469 PWH receiving care at the two clinics, 103 (7%) had confirmed SARS-CoV-2 infection. The median age was 56 (interquartile range [IQR] 45–62) years—72 PWH (69.9%)

**Table 1. Demographics and clinical characteristics of a cohort of PWH and COVID-19 by hospitalization status: 1/21/2020-1/20/2021.**

| | Total (n = 103) | Ambulatory (n = 69) | Hospitalized (n = 34) | p-value | OR (95% CI) | aOR (95%CI)* |
|---|---|---|---|---|---|---|
| Median age (IQR) | 56 (45–62) | 55 (42.5–59) | 58 (52.3–67.3) | 0.40# | | |
| Age≥50 yrs, n (%) | 72 (69.9) | 45 (65.2) | 27 (79.4) | 0.14 | | |
| Age>65 yrs, n (%) | 18 (17.5) | 8 (11.6) | 10 (29.4) | **0.025** | **3.18 (1.12–9.01)** | |
| Women, n (%) | 48 (46.6) | 32 (46.4) | 16 (47.1) | 0.95 | | |
| African Americans, n (%) | 45 (43.7) | 30 (43.5) | 15 (44.1) | 0.95 | | |
| Latinx, n (%) | 17 (16.5) | 13 (18.8) | 4 (11.8) | 0.27 | | |
| Median years living with HIV (IQR) | 16.5 (9–23.8) | 17 (8–25) | 16 (10–23) | 0.83# | | |
| History of AIDS, n (%) | 46 (44.7) | 30 (43.5) | 16 (47.1) | 0.75 | | |
| History of CD4<200 cells/mm$^3$, n (%) | 43 (41.8) | 28 (40.5) | 15 (44.1) | 0.77 | | |
| History of OI, n (%) | 24 (23.3) | 18 (26.1) | 6 (17.6) | 0.25 | | |
| On ART, n (%) | 101 (98.1) | 67 (97.1) | 34 (100) | 0.32 | | |
| Median CD4 count (IQR) | 735 (434–928) | 718 (397–931) | 784 (466–924) | 0.67# | | |
| Proportion VL suppressed (VL<200), n (%) | 95 (92.2) | 65 (95.2) | 30 (88.2) | 0.25 | | |
| Active smoking, n (%) | 23 (22.3) | 15 (21.7) | 8 (23.5) | 0.84 | | |
| Former smoking, n (%) | 58 (56.3) | 37 (53.6) | 21 (61.8) | 0.43 | | |
| Active substance use, n (%) | 14 (13.6) | 6 (8.7) | 8 (23.5) | **0.039** | **3.2 (1.0–10.2)** | |
| Active alcohol use, n (%) | 28 (27.2) | 19 (27.5) | 9 (26.5) | 0.91 | | |
| Diabetes mellitus, n (%) | 28 (27.2) | 14 (20.3) | 14 (41.2) | **0.025** | **2.75 (1.1–6.8)** | |
| Median HgbA1C (IQR) (n = 50) | 7.8 (6.4–8.5) | 7.1 (6.2–8.3) | 8.3 (6.6–9.0) | 0.20# | | |
| Chronic lung disease, n (%) | 35 (34) | 18 (26.1) | 17 (50) | **0.016** | **2.83 (1.20–6.7)** | **3.35 (1.28–8.72)** |
| Chronic kidney disease, n (%) | 23 (22.3) | 10 (14.5) | 13 (38.2) | **0.007** | **3.65 (1.39–9.57)** | |
| Cardiovascular disease, n (%) | 28 (27.2) | 13 (18.8) | 15 (44.1) | **0.007** | **3.4 (1.37–8.42)** | **3.4 (1.27–9.12)** |
| Hypertension, n (%) | 53 (51.5) | 31 (44.9) | 22 (64.7) | 0.059 | 2.25 (0.96–5.25) | |
| Hepatitis C, n (%) | 23 (22.3) | 11 (15.9) | 12 (35.3) | **0.027** | **2.88 (1.1–7.47)** | |
| Median BMI (IQR) | 28.9 (25.2–34.9) | 29.3 (25.7–35.0) | 26.8 (24.7–36.0) | 0.30# | | |
| Obesity (BMI>30), n (%) | 45 (43.7) | 32 (46.4) | 13 (38.2) | 0.43 | | |
| Any co-morbidity, n (%) | 93 (90.3) | 62 (89.9) | 31 (91.2) | 0.83 | | |
| Number of co-morbidities | 3 (1–4) | 2 (1–3) | 4 (2–5) | **0.001*** | **1.61 (1.22–2.13)** | |

*Logistic regression

#Mann Whitney U; PWH-persons with HIV; BMI-Body mass index

were >50 years old, while 17.5% were >65 years old), 46.6% were women, 43.7% were African Americans, and 16.5% were Latinx (Table 1). The median CD4 count was 735 cells/mm$^3$ (IQR 434–928), 98.1% of the patients were prescribed ART, and 92.2% had HIVVL suppression. Among the cohort, 88% had co-morbidities, 51.5% had hypertension; 43.7%, obesity; 34%, chronic lung disease; 27.2%, cardiovascular disease; 27.2%, diabetes mellitus; and 22.3%, chronic kidney disease. While 56.3% of PWH were former smokers, active use of tobacco was documented in 22.3%, alcohol in 27.2%, and illicit substances in 13.6%.

## Comparison of hospitalized vs ambulatory PWH with COVID-19

Hospitalized PWH represented less than 1% of all COVID-19-related admissions to YNHH during the study period. Thirty-four PWH (33.0%) were hospitalized and 69 (67.0%) were managed in the ambulatory setting. In the bivariate analysis, age and multiple co-morbidities such as diabetes, chronic lung disease, chronic kidney disease, and cardiovascular disease, were associated with increased hospitalization (p<0.05). However, in the adjusted analysis (Table 1), those who were hospitalized were more likely to be at least 65 years old (OR: 3.11,

**Table 2. Characteristics of hospitalized patients.**

| HIV clinical parameters | |
|---|---|
| Median Temperature on admission, degrees C (IQR) | 37.5 (36.8–38.2) |
| Median CD4 Count cells/mm3 during hospitalization (n = 15) | 417.8 (186–617) |
| HIV Viral Suppression during hospitalization (<200 copies/mm3) (n = 20) | 19 (95%) |
| **COVID therapy** | |
| Hydroxychloroquine | 16 (47.1%) |
| Tocilizumab | 8 (23.5%) |
| Remdesivir | 5 (14.7%) |
| Methylprednisolone | 5 (14.7%) |
| Atazanavir | 3 (8.8%) |
| Lopinavir/ritonavir | 2 (5.9%) |
| Convalescent plasma | 0 (0%) |
| No therapy | 6 (17.6%) |
| **Clinical outcomes** | |
| Length of Stay (days) | 9 (3–15) |
| Escalation of care to critical care unit | 11(32.4%) |
| Stepdown: | 6 (17.6%) |
| ICU: | 5 (14.7%) |
| Mechanical ventilation | 4 (11.8%) |
| Mortality (of hospitalized/all PWH) | 1 (2.94%/0.97%) |

PWH-persons with HIV

95% CI: 0.97–9.98) and to have chronic lung disease (OR: 3.35, 95% CI: 1.28–8.72) or cardiovascular disease (OR: 3.4, 95% CI: 1.27–9.12). Moreover, incremental numbers of co-morbidities were associated with hospitalization (OR: 1.61, 95% CI: 1.22–2.13). AIDS history and last CD4 count <200 cells/mm$^3$ were not associated with hospitalization. There was no significant difference in ART coverage or HIVVL suppression between inpatients and outpatients.

**Clinical course of hospitalized patients.** Among those who were hospitalized (Table 2), the most frequently prescribed inpatient therapies for COVID-19 were hydroxychloroquine (47.1%), tocilizumab (23.5%), and remdesivir and steroids (14.7%). A small proportion (14.7%) received lopinavir/ritonavir or atazanavir. During hospitalization, five patients (14.7%) required escalation to ICU level of care, six patients (17.6%) needed intermediate (step-down) care, and four (11.8%) had to be placed on mechanical ventilation. The median length of stay was nine days (IQR 3–15). Only one patient (2.9%) died within 30 days of hospital admission. In a subset of hospitalized patients (n = 15), repeat CD4 counts revealed a decreased median of 417.8 (IQR 186–617) cells/mm$^3$. Repeat HIVVL testing done for 59% of patients showed full viral suppression in 95% of that subset (Table 2).

## Discussion

This study describes the impact of COVID-19 infection on PWH during the first year of the pandemic at a single urban academic medical center in the US and identifies correlates of hospitalization. Hospitalization is associated with host factors, including older age and individual comorbidities; increasing burden of comorbidities resulted in higher risk of hospitalization.

The predominant co-morbidities for hospitalized patients—chronic lung and cardiovascular disease—were similar to those for patients in other COVID-19 cohorts [12, 17–19]. While other studies have demonstrated that diabetes and chronic kidney disease were significantly

associated with severe disease and/or poor outcomes, we found that active substance use and Hepatitis C, were significantly correlated on bivariate but not on adjusted analysis, possibly attributable to sample size [4]. Similar to other studies, older age (>65yo) was independently associated with hospitalization. The median age of the cohort was similar to that in other studies and was lower than that of hospitalized HIV-uninfected patients with COVID-19 [8, 12, 17].

With the benefit of ART effectiveness, PWH are living longer, but are increasingly being diagnosed with new and multiple co-morbidities, particularly among communities of color [20]. While there is growing evidence that HIV-associated immunosuppression is not thought to be associated with COVID-19-related hospitalization or death, indirect measures of HIV and aging, as manifested in co-morbidities, still correlate with COVID-19 hospitalization. More recent reports from Moran et al. [21] which were conducted in a similar timeframe as our study, also found that co-morbidity burden was associated with hospitalization in a dose-dependent fashion when comparing hospitalized and nonhospitalized PWH with COVID-19. Sun et al. [22], in a National COVID Cohort Collaborative (N3C) study, looked at a large database of PWH with COVID-19 at US academic medical centers, compared with solid organ transplant patients, again noted that increased hospitalization and mechanical ventilation among PWH was related to the co-morbidity burden and not to specific demographics. However, the odds of hospitalization were observed to be higher for those with HIV compared to those HIV-uninfected.

Despite well-maintained pre-COVID CD4 counts and suppressed HIVVL, HIV-associated immune dysfunction may affect the person's ability to respond to the exuberant inflammatory reaction, resulting in clinical deterioration from COVID-19 [23]. Yendewa et al. [24] assessed a large US health-care network of 44 facilities where PWH under less intensive ART and with much lower rates of viral suppression had higher odds of hospitalization, ICU admission, and mechanical ventilation, than patients without HIV infection. Co-morbidities were not reported, but the two groups of patients did have comparable 30-day mortality rates. Interestingly, among a subset of our hospitalized patients with CD4 counts performed after admission in this study, there was a significant drop in the absolute CD4 count consistent with nonspecific changes seen in acute illness.

The direct antiviral benefit of ART could be another explanation for less severe disease. A lower risk of hospitalization and severe disease was observed among other cohorts of patients on tenofovir disoproxil [14, 25]. The current study, though limited by sample size, showed no indication that administration of ART had any effect on the clinical course or outcome. Differences in outcomes such as mortality may also have been affected by the variety of care standards used in each unique facility. Maintaining ART resulted in favorable outcomes for the overwhelming majority of patients.

## Limitations

We recognize several limitations in this study. First, this was a retrospective review of medical records and missing data may introduce bias. Second, the study was conducted at a single urban academic medical center, and so may not be representative of all PWH. Third, hospitalized cases were identified through a health systems database with laboratory confirmation, leaving open the possibility that there may have been PWH who received their HIV care at our institution but were tested for SARS-CoV-2 or hospitalized elsewhere resulting in an underestimate of the COVID-19 burden among PWH. Fourth, although reports show that COVID-19 disproportionately affects disadvantaged populations [26], the PWH cohort in the current study, consisting largely of minorities and people over 50 years of age with multiple co-morbid

conditions achieved overall favorable outcomes, possibly due to ready access to advanced treatments within an academic health center. Next, the PWH in this sample had high CD4 counts and more than 90% HIVVL suppression and may not be generalizable to the global population of PWH. Finally, these data reflect the COVID-19 trends among PWH before the initiation of mass vaccination and the clinical emergence of the delta variant in the US. Vaccination and new SARS-CoV-2 variants may alter risk factors for hospitalization and severe disease [27].

## Conclusion

In this retrospective study, we identified 103 PWH coinfected with COVID-19 in an urban US setting who were hospitalized over the first year of the pandemic. Compared with PWH with COVID-19 managed as outpatients, those needing hospitalization were found to be older and have multiple non-communicable disease comorbidities though mortality was low. Notably, increased numbers of comorbidities increased the risk of hospitalization. HIV-attributable factors were not associated with hospitalization for COVID-19.

## Supporting information

**S1 Table. Demographics and clinical characteristics of PWH with SARS-CoV-2, 1/21/20-1/20/21.**
(PDF)

**S2 Table. Characteristics of hospitalized PWH.**
(PDF)

## Acknowledgments

We are we grateful for the editing assistance of Mary-Ann Asico in the preparation of the report. Most importantly, we thank our patients.

## Author Contributions

**Conceptualization:** Michael D. Virata, Sheela V. Shenoi, Merceditas S. Villanueva, Lydia A. Barakat.

**Data curation:** Michael D. Virata, Sheela V. Shenoi, Joseph Ladines-Lim, Lydia A. Barakat.

**Formal analysis:** Michael D. Virata, Sheela V. Shenoi, Merceditas S. Villanueva, Lydia A. Barakat.

**Investigation:** Michael D. Virata, Lydia A. Barakat.

**Methodology:** Michael D. Virata, Sheela V. Shenoi, Merceditas S. Villanueva, Lydia A. Barakat.

**Project administration:** Michael D. Virata.

**Supervision:** Michael D. Virata.

**Visualization:** Michael D. Virata.

**Writing – original draft:** Michael D. Virata, Sheela V. Shenoi, Merceditas S. Villanueva, Lydia A. Barakat.

**Writing – review & editing:** Michael D. Virata, Sheela V. Shenoi, Joseph Ladines-Lim, Merceditas S. Villanueva, Lydia A. Barakat.

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
