## [Decision Letter · Decision Letter 0]

13 Sep 2021

PONE-D-21-24226Increasing Numbers of Non-communicable Disease Co-morbidities: Major Risk Factors for Hospitalization among a Cohort of People with HIV and  COVID-19 CoinfectionPLOS ONE

Dear Dr. Virata,

Thank you for submitting your manuscript to PLOS ONE. After careful consideration, we feel that it has merit but does not fully meet PLOS ONE’s publication criteria as it currently stands. Therefore, we invite you to submit a revised version of the manuscript that addresses the minor points raised during the review process.

We look forward to receiving your revised manuscript.

Kind regards,

Manish Sagar, MD

Academic Editor

PLOS ONE

Journal Requirements:

Reviewers' comments:

Reviewer's Responses to Questions

**Comments to the Author**

1. Is the manuscript technically sound, and do the data support the conclusions?

Reviewer #1: Yes

Reviewer #2: Yes

2. Has the statistical analysis been performed appropriately and rigorously? 

Reviewer #1: Yes

Reviewer #2: Yes

3. Have the authors made all data underlying the findings in their manuscript fully available?

Reviewer #1: Yes

Reviewer #2: Yes

4. Is the manuscript presented in an intelligible fashion and written in standard English?

Reviewer #1: No

Reviewer #2: Yes

5. Review Comments to the Author

Reviewer #1: The work describes Increasing Numbers of Non-communicable Disease Co-morbidities: ‎Major Risk Factors for Hospitalization among a Cohort of People with HIV and COVID-19 ‎Coinfection. The manuscript is well organized and addresses the topic sufficiently. Tables ‎are suitable for description. The authors are suggested to apply the following comments in ‎the manuscript:‎

‎1.‎ Importantly, the authors need to significantly improve the writing.‎

‎2.‎ Keywords are not standard. Please match the keywords with the mesh.‎

‎3.‎ The addressed problem is not clear. Give more emphasis on the problems and the gap ‎you intend to fill. ‎

‎4.‎ Refer to the following articles in the introduction:‎

https://pubmed.ncbi.nlm.nih.gov/32556781/‎

https://pubmed.ncbi.nlm.nih.gov/33101547/‎

https://pubmed.ncbi.nlm.nih.gov/33363000/‎

https://pubmed.ncbi.nlm.nih.gov/33357637/‎

‎5.‎ The journal framework is not followed. ‎

‎6.‎ The limitations of the research are not mentioned.‎

‎7.‎ In the discussion section, you should fully discuss the study findings with the findings ‎of other related studies. In this section, you can use the following resources to enrich ‎the content of the discussion.‎

https://pubmed.ncbi.nlm.nih.gov/34103090/‎

https://pubmed.ncbi.nlm.nih.gov/34016183/‎

https://pubmed.ncbi.nlm.nih.gov/34217366/‎

Reviewer #2: I think this work adds to the growing body of literature supporting that HIV in of itself is not a driver of poor Covid-19 outcomes among people with HIV. 2 things I would suggest:

- can you specifically comment on the renal status of this cohort? Renal disease is more common among PWH and has been associated with poor Covid-19 outcomes.

- I recommend changing the title, it's a bit confusing

6. PLOS authors have the option to publish the peer review history of their article (what does this mean?). If published, this will include your full peer review and any attached files.

Reviewer #1: No

Reviewer #2: **Yes: **Richard Jason Silvera

---

## [Author Response · Author response to Decision Letter 0]

3 Nov 2021

November 1, 2021

Manish Sagar MD

Academic Editor

Plos One

Dear Dr. Sagar,

On behalf of all the authors, we are thankful for the careful review of our manuscript “Increasing Numbers of Non-communicable Disease Comorbidities Increases Risk for Hospitalization among People with HIV and COVID-19.” 

We have now revised the manuscript incorporating all the reviewers’ feedback and below include a point by point response to the reviewers’ comments. We believe the manuscript has been strengthened as a result of this review and hope that you now find it suitable for publication. 

Please do not hesitate to let us know if there are any questions. We look forward to your final decision.

Sincerely

Michael D. Virata, MD

We thank the reviewers for their thorough and constructive critique. 

Reviewer's Responses to Questions:

Comments to the Author  

1. Is the manuscript technically sound, and do the data support the conclusions?  The manuscript must describe a technically sound piece of scientific research with data that supports the conclusions. Experiments must have been conducted rigorously, with appropriate controls, replication, and sample sizes. The conclusions must be drawn appropriately based on the data presented. 

Reviewer #1: Yes

Reviewer #2: Yes

Response: We thank the reviewers for the positive feedback

2. Has the statistical analysis been performed appropriately and rigorously?

Reviewer #1: Yes

Reviewer #2: Yes

Response: We appreciate the reviewer’s acceptance of our statistical analysis

3. Have the authors made all data underlying the findings in their manuscript fully available?  The PLOS Data policy requires authors to make all data underlying the findings described in their manuscript fully available without restriction, with rare exception (please refer to the Data Availability Statement in the manuscript PDF file). The data should be provided as part of the manuscript or its supporting information or deposited to a public repository. For example, in addition to summary statistics, the data points behind means, medians and variance measures should be available. If there are restrictions on publicly sharing data—e.g. participant privacy or use of data from a third party—those must be specified.

Reviewer #1: Yes

Reviewer #2: Yes

Response: We thank the reviewer for their finding our data are appropriately displayed, available and accurate

4. Is the manuscript presented in an intelligible fashion and written in standard English?  PLOS ONE does not copyedit accepted manuscripts, so the language in submitted articles must be clear, correct, and unambiguous. Any typographical or grammatical errors should be corrected at revision, so please note any specific errors here.

Reviewer #1: No

Reviewer #2: Yes

Response: We appreciate the constructive feedback provided by the reviewers. We have made significant changes in the writing of the manuscript to make it more suitable for publication as highlighted in the attached revision. In addition, please refer to our detailed and point by point response below.

Reviewer #1: The work describes Increasing Numbers of Non-communicable Disease Co-morbidities: ‎Major Risk Factors for Hospitalization among a Cohort of People with HIV and COVID-19 ‎Coinfection. The manuscript is well organized and addresses the topic sufficiently. Tables ‎are suitable for description. The authors are suggested to apply the following comments in ‎the manuscript:‎ ‎

1.‎ Importantly, the authors need to significantly improve the writing.‎ :

Response: Thank you for this suggestion. We have substantially revised the manuscript to improve the writing and enhance the readability as highlighted in the attached manuscript.

2.‎ Keywords are not standard. Please match the keywords with the mesh.

Response: Thank you for this comment. We have clarified the keywords in accordance with the journal instructions and hope these are appropriate. We will seek additional guidance from the editorial team.

3.‎ The addressed problem is not clear. Give more emphasis on the problems and the gap ‎you intend to fill. ‎ ‎

Response: Thank you for this feedback. We have revised the introduction to directly state the gap that we are addressing in this manuscript as reflected in the attached updated version. 

4.‎ Refer to the following articles in the introduction

https://pubmed.ncbi.nlm.nih.gov/32556781/‎

https://pubmed.ncbi.nlm.nih.gov/33101547/‎

https://pubmed.ncbi.nlm.nih.gov/33363000/‎

https://pubmed.ncbi.nlm.nih.gov/33357637/‎

‎Response: Thank you for these suggested references. We have selected and incorporated the reference on the predictors of mortality in patients with COVID-19---a systematic review into the introduction . Although we learned from reviewing the other references, we did not include all of them since they were not directly related to our study. We look forward to optimizing the manuscript with further clarification from the reviewer. 

5.‎ The journal framework is not followed. ‎

Response: We appreciate this feedback. We have reviewed and revised the font of the headings, in line with journal instructions. 

‎6.‎ The limitations of the research are not mentioned.‎

Response: Thank you for this feedback. We have emphasized the limitations in the discussion section of the study. 

‎7.‎ In the discussion section, you should fully discuss the study findings with the findings ‎of other related studies. In this section, you can use the following resources to enrich ‎the content of the discussion.‎

https://pubmed.ncbi.nlm.nih.gov/34103090/‎

https://pubmed.ncbi.nlm.nih.gov/34016183/‎

https://pubmed.ncbi.nlm.nih.gov/34217366/‎

Response: Thank you for these suggested references. We have incorporated the article reviewing COVID-19 variants. However, the articles discussing genetic susceptibility and impact of COVID-19 prevention measures on other infections are both beyond the scope of our study. 

Reviewer #2: I think this work adds to the growing body of literature supporting that HIV in of itself is not a driver of poor Covid-19 outcomes among people with HIV. 2 things I would suggest: 

1. can you specifically comment on the renal status of this cohort? Renal disease is more common among PWH and has been associated with poor Covid-19 outcomes. 

Response: Thank you for this important question. Chronic kidney disease was prevalent in our patient sample and significantly associated with hospitalization for COVID-19, though was not an independent correlate. We have now stated this in the discussion to address this comment.

2. I recommend changing the title, it's a bit confusing

Response: Thank you for this feedback. We agree that the title could be improved, and we have changed the title to make it more concise.

---

## [Editor Report · Decision Letter 1]

8 Nov 2021

Cumulative burden of non-communicable diseases predicts COVID hospitalization  among people with HIV:   A one-year retrospective cohort study

PONE-D-21-24226R1

Dear Dr. Virata,

We’re pleased to inform you that your manuscript has been judged scientifically suitable for publication and will be formally accepted for publication once it meets all outstanding technical requirements.

Kind regards,

Manish Sagar, MD

Academic Editor

PLOS ONE
---

## [Editor Report · Acceptance letter]

18 Nov 2021

PONE-D-21-24226R1 

Cumulative burden of non-communicable diseases predicts COVID hospitalization  among people with HIV:   A one-year retrospective cohort study 

Dear Dr. Virata:

I'm pleased to inform you that your manuscript has been deemed suitable for publication in PLOS ONE. Congratulations! Your manuscript is now with our production department. 

Kind regards, 

on behalf of

Dr. Manish Sagar 

Academic Editor

PLOS ONE